# Outcomes of Pregnancy in Sickle Cell Disease Patients: Results from the Prospective ESCORT-HU Cohort Study

**DOI:** 10.3390/biomedicines11020597

**Published:** 2023-02-17

**Authors:** Anoosha Habibi, Giovanna Cannas, Pablo Bartolucci, Ersi Voskaridou, Laure Joseph, Emmanuelle Bernit, Justine Gellen-Dautremer, Corine Charneau, Stephanie Ngo, Frédéric Galactéros

**Affiliations:** 1Sickle Cell Referral Center, Internal Medicine Unit, Henri Mondor Hospital, Assistance Publique-Hôpitaux de Paris, U-PEC, 94000 Créteil, France; 2INSERM-U955, Institut Mondor, Université Paris-Est Créteil, Team 2 Transfusion et Maladies du Globule Rouge, Laboratoire d’Excellence GR-Ex, 94000 Créteil, France; 3Edouard Herriot Hospital, 69003 Lyon, France; 4Thalassemia and Sickle Cell Disease Center, “Laiko” General Hospital, 115 27 Athens, Greece; 5Biotherapy Department, Necker Children’s Hospital, Assistance Publique-Hôpitaux de Paris, 75610 Paris, France; 6Sickle Cell Referral Center, CHU Guadeloupe-Pôle Parents-Enfants—Hôpital Ricou, BP465, Pointe à Pitre, CEDEX, 97159 Guadeloupe, France; 7CHU la Milétrie, 86000 Poitiers, France; 8Basse-Terre Hospital, Basse-Terre 97109, Guadeloupe; 9Delafontaine Hospital, 93200 Saint-Denis, France

**Keywords:** hydroxyurea, sickle cell disease, patients, pregnancy

## Abstract

Sickle cell disease (SCD) refers to a group of inherited hemoglobin disorders in which sickle red blood cells display altered deformability, leading to a significant burden of acute and chronic complications, such as vaso-occlusive pain crises (VOCs). Hydroxyurea is a major therapeutic agent in adult and pediatric sickle cell patients. This treatment is an alternative to transfusion in some complications. Indeed, it increases hemoglobin F and has an action on the endothelial adhesion of red blood cells, leukocytes, and platelets. Although the safety profile of hydroxyurea (HU) in patients with sickle cell disease has been well established, the existing literature on HU exposure during pregnancy is limited and incomplete. Pregnancy in women with SCD has been identified as a high risk for the mother and fetus due to the increased incidence of maternal and non-fetal complications in various studies and reports. For women on hydroxyurea at the time of pregnancy, transfusion therapy should probably be initiated after pregnancy. In addition, there is still a significant lack of knowledge about the incidence of pregnancy, fetal and maternal outcomes, and management of pregnant women with SCD, making it difficult to advise women or clinicians on outcomes and best practices. Therefore, the objective of this study was to describe pregnancy outcomes (*n* = 128) reported in the noninterventional European Sickle Cell Disease COhoRT-HydroxyUrea (ES-CORT-HU) study. We believe that our results are important and relevant enough to be shared with the scientific community.

## 1. Introduction

Sickle cell disease (SCD) is a recessive genetic disease related to the production of abnormal hemoglobin (HbS) [1,2]. Sickle red blood cells present altered deformability, become fragile, and are thus more prone to lysis, thereby resulting in chronic hemolytic anemia [1,2]. This may lead to vaso-occlusive pain crises (VOCs), ischemia, and inflammation with acute and chronic lesions.

SCD represents the most common monogenic disorder in the world and it affects millions of patients living in sub-Saharan Africa and the Indian sub-continent [3]. Previous results have shown that genetic changes include a homozygous missense mutation [Glu6Val, rs334, replacement of normal glutamic acid with valine] in the β-globin gene [4]. This modification in a single DNA base leads to a cascade of physiological consequences that can impact numerous organs and systems. In low-oxygen conditions, the polymerization of the sickle β-globin leads to erythrocytes displaying a crescent or sickled shape, thus the designation ‘sickle cell’ disease [5]. It is of note that reduced levels of hemoglobin and low oxygen affinity have been shown to further exacerbate hemoglobin S polymerization and sickle cell formation [6]. Reciprocally, hemoglobin S polymerization itself was also evidenced to reduce the affinity for oxygen and to stabilize the deoxygenated form of the molecule. In addition to a change in shape, sickling results in the loss of potassium, and a gain in intracellular sodium and calcium in the sickled erythrocytes [7]. Hence, sickle red blood cells present altered deformability, become fragile, and are more prone to lysis [5]. Together, these factors may lead to hemolytic anemia and to vaso-occlusion in the small blood vessels, which causes most of the other clinical features, including acute painful crises [2]. Indeed, other complications related to sickle cell disease have been evidenced and include stroke, pulmonary hypertension, renal dysfunction, retinal disease, leg ulcers, cholelithiasis, avascular necrosis (which commonly affects the femoral head and may necessitate hip replacement), and inflammation with acute and chronic lesions [1,2].

Pregnancy in SCD patients is considered a risk both for the mother and the fetus. The management of sickle cell pregnancy thus represents a real challenge for clinicians. Indeed, pregnancy was shown to be associated with exacerbation of sickle cell disease and may place women, especially those with sickle cell anemia (HbSS), at an increased risk for obstetric complications. Retrospective data from limited numbers of patients in the 1960s and 1970s suggested an augmented rate of maternal mortality in sickle cell disease when compared to that of women without the disease [8]. In the same vein, some obstetric complications have been observed, including placental abruption, pre-eclampsia, toxemia, and preterm labor, which have been reported more frequently in women with HbSS than in healthy African American women without sickle cell anemia [9]. An increased rate of vaso-occlusion crises (VOCs), infections, acute chest syndromes (ACS) and thromboembolic events was also described [1]. This augmented susceptibility to vaso-occlusion may account for the areas of fibrosis, villous necrosis, and infarction observed in the placenta from pregnancies in women with SCD [10]. Notably, all of this may then be associated with chronic fetal hypoxia and adverse fetal outcomes such as prematurity and fetal death [11,12]. Nevertheless, there is still a gap of knowledge about the incidence of pregnancy and associated fetal and maternal outcomes, as the few existing publications related to HU exposure during pregnancy have mostly involved small number of patients [13].

Hydroxycarbamide or hydroxyurea (HU) has been approved since 2007 in the European Union (EU) for the prevention of recurrent painful VOC, including acute chest syndromes, in adults, adolescents and children older than 2 years old suffering from SCD. Since 2017, HU h approved in the United States of America (USA) to reduce the frequency of painful crises and the need for blood transfusions in pediatric patients from 2 years of age with sickle cell anemia with recurrent moderate to severe painful crises. Notably, among the most relevant publications that had been recognized in the field, Montalembert et al. has particularly proved the efficacity of this treatment in sickle cell disease patients, where a significant decrease in the number of vaso-occlusive crises could be noted [14,15]. Mechanistically, HU was shown to be able to induce the production of hemoglobin F through the recruitment of a population of erythroid precursors that can continue γ-chain synthesis [16]. Only a few cases of HU exposure during pregnancy in women with SCD have been published [17,18,19,20]. Importantly, as no malformation in newborns was reported due to HU treatment to date, it was suggested that the risk of deleterious teratogenic effects shown in some animal species, such as mice and monkeys, at the doses comparable to human doses (in mg/m^2^) may have been overestimated in humans [21,22,23]. Given the limited literature on HU exposure during pregnancy and the potential impact of HU in this population, patients treated with HU and wishing to conceive should stop the treatment 3 to 6 months before pregnancy in EU, and at least 6 months before pregnancy in the United States of America (USA). There is a substantial need for additional studies of HU exposure during pregnancy in SCD women. In this study, we present the results of pregnancy outcomes reported in the European Sickle Cell Disease COhoRT-HydroxyUrea (ESCORT-HU) study. We consider the data that emanate from ESCORT-HU therapeutically significant in order to improve the management of HU treatment before and during pregnancies in real-life of women who present with a severe form of SCD.

## 2. Materials and Methods

ESCORT-HU was designed as a multicentric, prospective, non-interventional, European cohort study initiated to collect information on the long-term safety of HU tablets when they are used in current practice. Patients were enrolled in the study cohort if they satisfied the following criteria, including: male or female ambulatory patients, aged 2 years and more, with symptomatic SCD, entitled to be treated (patients enrolled under protocol Amendment 1) or treated (patients enrolled under protocol Amendment 2) with HU tablets, having been informed of the current study by the initiating physician and consenting to participate in the cohort (for children, the persons having parental authority having been informed and having given consent for participation). No exclusion criteria were defined in the study. There were no obligatory visits for the study. Importantly, visits were based on those planned by the clinicians as part of their routine practice. The population gave their consent or non-opposition (depending on local regulations) to participate in the study. This study was approved by National Ethics Committees and lasted for 10 years from January 2009 to March 2019.

To the best of our knowledge, this is the first study in which all pregnancy cases were collected irrespective of action taken with HU treatment before or during pregnancy. As in the Multicenter Study of Hydroxyurea in SCD (MSH) clinical trial, data were collected from routine visits of the patients to hospital. Clinical data from medical records through electronic case report forms (eCRFs) were collected by a contract research organization and were monitored in accordance with the established protocol. Additionally, quality control of the displayed data has been performed since 2012. 

At baseline (during the previous 12 months) and during the follow-up visits, clinical data and laboratory parameters have been collected, including acute SCD complications, such as VOC lasting ≥48 h and requiring hospitalization, ACS, number and duration of hospitalizations, number of blood transfusions, hemoglobin (Hb) level, and finally fetal hemoglobin (HbF %).

Statistical analysis was performed using SAS^®^ software version 9.4 (SAS Institute Inc., Cary, NC, USA) on Windows Server. For quantitative variables, we used the following: number of observations, number of missing observations, mean, standard deviation, median, and first and third quartile were used; for categorical variables, we used: number of observations in each population, number of observations, and percentage for each category. The number of missing observations was included in the following tables and the displayed percentages were calculated by omitting the missing observations. 

Importantly, one of the main purposes of collecting all pregnancy cases in ESCORT-HU is to improve the management of HU treatment before and during pregnancies in real-life women with severe SCD.

## 3. Results

A total of 1906 patients with sickle cell disease were involved in the study at sixty-three centers in France, Germany, Greece, and Italy. The gender distribution in the cohort was 854 men (45%) and 1052 women (55%). The mean duration of HU treatment was 42.4 months (SD, 23.8) for HU-naïve patients at inclusion in the study (*n* = 976) and 122.8 months (SD, 62.9) for patients who had already received HU before inclusion in the study (*n* = 926); 51 patients (2.6%) were lost during the follow-up (Figure 1) [15]. As displayed hereafter, the most common hemoglobin genotypes were HbSS in 84.7% and Hb Sβ+ thalassemia in 7% of cases. The follow-up median duration was 45 months for a total number of 7309 patient-years of observation. 

At the time of enrollment, more than two-thirds of the women were pubescent (*n* = 704, 68.3%) and therefore they were considered to be of childbearing age. The median age of the first mensestruation was 15 [13–19] years. Forty-eight girls reached puberty during the study, but none of them became pregnant. Among the women, 374 women (54.8%) were previously pregnant, with 345 having at least one child. The majority of women (65.7%) for whom data were available (*n* = 513) were not using any method of contraception. The number of pregnancies before and during the study is shown in Table 1.

During the study, 125 pregnancies occurred in 101 women and there were 128 births, including three twins. The women’s characteristics are displayed in Table 2. Interestingly, the majority of these women were SS homozygotes but we had one patient with SC and four patients with Sβ+thalassemia, these two genotypes being rarely treated with HU. Nineteen women were found to be pregnant twice during the follow-up period and five reported three pregnancies. A first pregnancy during the follow-up was reported in 38.6% of women at a mean age of 27.2 + 4.7 years old (SD). Additionally, the median age of the first pregnancy was 30 [26–33] years and the Hb level of Hb was 8.8 [7.9–9.5] g/dL. Notably, the first indication of HU in these SCD patients was to treat vaso-occlusive crises (VOCs) and in second place acute chest syndrome (ACS). 

A total of 110 pregnancies (*n* = 91 women) were potentially exposed to HU, as HU was not stopped at least for 15 days before conception, and 15 pregnancies (*n* = 10 women) were not exposed to HU. The periods of exposure to HU are displayed in Table 3; globally 85% of the pregnancies had been exposed to HU during the first trimester, corresponding to the period of teratogenicity when it is better to circumvent its exposure. The mean dose of HU received during exposed pregnancy in SCD women was 16.8 mg/kg/d (SD, 4.8) (*n* = 80).

Patients had a median of 10 [8–12] visits during their follow-up period with a median of 52.2 [37.2–60.1] months. Concerning the biological parameters, when available, the mean level of Hb in the year prior to the beginning of pregnancy was 8.8 + 1 g/dL (SD) (*n* = 56) and the mean level of HbF was 10.9% + 7.8 (SD) (*n* = 36). Patients reported a median [min-max] of 0 [0–6] VOC (*n* = 66), 0 [0–3] ACS (*n* = 66), and 0 [0–6] hospitalizations (*n* = 66) lasting 0 [0–50] days (*n* = 66).

Pregnancy outcomes (*n* = 128) are presented according to HU exposure (*n* = 110) or not (*n* = 15) during pregnancy in Table 4. Spontaneous abortions were separated into miscarriages (*n* = 20) (gestational age (GA) < 24 weeks) and stillbirths (*n* = 1) (GA ≥ 24 weeks). One therapeutic abortion was performed due to maternal cardiomyopathy, one stillbirth was reported during the study without additional data available and without suspected link to HU, and seven outcomes remained unknown due to loss of the patients’ follow-up.

In four pregnancies, HU was not stopped at any time. Indeed, these four women had severe symptoms of SCD, including VOC, severe chronic anemia, heart disease, or post-transfusion alloimmunization, leading to the maintenance of HU treatment. Among these exposed patients, one of them was hospitalized during pregnancy for preeclampsia and another one for internal hypogastric artery hemorrhage and premature rupture of membranes. Notably, none of these events was suspected to be related to the treatment with HU. These four pregnancies resulted in two normal births (GA ≥ 36 weeks) and two preterm births (GA < 36 weeks).

Among the 110 exposed pregnancies, 60 maternal complications were reported in 41 pregnancies in 37 women with sickle cell disease. No details were available regarding the time between HU exposure and/or HU discontinuation and the occurrence of the complications, except for three women who developed VOC or ACS after HU discontinuation. The 60 maternal complications are shown in Table 5. None of these single or multiple complications were suspected to be related to HU. Transfusions were also reported in 43 pregnancies and caesarean sections in 37 pregnancies. Although SCD women treated with HU usually had severe symptoms of the underlying disease and were at particular risk, no maternal death was reported during the study. Among the 15 pregnancies without HU exposure, three complications were reported in the same pregnancy after HU discontinuation, including one painful VOC and two hospitalizations related to sickle cell disease (one lasting 5 days and the other one 4 days). The live birth rate was 73% for pregnancies with maternal HU exposure, excluding elective abortions. The mean newborn weight was 2575 g (SD, 511). Three intrauterine growth defects were reported, but again none of them were suspected to be related to the treatment with HU. Notably, the state of data collection per se does not allow for accurate detailing of complications or treatments received during the pregnancies.

During the study, 11 events in newborns were reported. The 10 reported events in neonates whose mothers had been exposed to HU included one case of fever with no evident cause and treated with antibiotics, one scimitar syndrome, one fetal arrhythmia, one cardiac disorder, one patent ductus arteriosus, one neonatal respiratory distress, one hyaline membrane disease, and three fetal growth retardations. The committee of clinical investigators studied these files and concluded on the imputability or not of the anomalies according to the period of exposure, duration of exposure, and the nature of the anomalies. However, none of these events were suspected to be related to HU.

## 4. Discussion

Sickle cell disease is a recessive genetic disease related to the production of abnormal hemoglobin (HbS) with sickle red blood cells presenting altered deformability, thereby leading to chronic hemolytic anemia and vaso-occlusive pain crises. In addition, it is associated with lifelong morbidity and reduced life expectancy. Major advances in the management of SCD have led to 99% survival in children suffering from SCD after the age of 5 years in France and, consequently, to an increasing number of women reaching childbearing age and being potentially willing to conceive [24]. In this study, we observed that the age of first menstruation in girls remains relatively late, around 15 years. This had already been described in sickle cell disease patients and these observations led to the conclusion that there does not seem to be any fertility problem in these patients, whether treated or not with HU. However, pregnancy in women with SCD remains a period of high maternal and fetal risk, especially in patients who were symptomatic enough to receive HU therapy—no maternal mortality could be found in our study. 

Most of the published studies dealing with pregnancy outcomes in sickle cell disease patients have been either retrospective, case reports, or undertaken with a small series of patients. Kroner et al. recently published the results of the study using the Sickle Cell Disease Implementation Consortium (SCDIC) Registry with 167 SCD women, with at least one pregnancy conceived while taking HU [20]. One important limitation is that the study was not designed for a pregnancy follow-up and the pregnancy history was self-reported by SCD women [20]. Therefore, the results that emanate from ESCORT-HU, which is a prospective and multicentric study conducted in several European countries, importantly and significantly contribute to improving knowledge about pregnancy outcomes in SCD women exposed to HU. This is one of the few studies showing the frequency of miscarriage in sickle cell patients. Similarly, it brings valuable new insights about the management of HU treatment in real life not only before but also during pregnancy.

Although women are advised to stop HU at least 3 months before pregnancy and restart it after the end of breastfeeding, the continuation of HU is often necessary in real clinical practice. Stopping this medication is indeed accompanied by crisis resumptions and/or other complications, and the only alternative remains a transfusion program. Publications dealing with the risks of transfusion, particularly delayed hemolytic transfusion reactions and their frequency during pregnancy, make this therapeutic option less feasible. Indeed, in a series of 99 delayed hemolytic transfusion reactions, 33% of patients were pregnant. In some countries, clinicians have tended to continue treatment until pregnancy is confirmed [25]. HU remains the only alternative treatment that protects both the mother and the fetus from vaso-occlusive complications in poly-alloimmunized patients or those with a history of delayed hemolytic transfusion reactions. Recently, Montironi et al. recommended that all SCD pregnancies be treated not only depending on the treatment previously adopted, but also based on the patient’s history of sickle cell disease [26]. This study also proposed HU for severely and mildly affected patients, respectively, from the second and third trimesters [26]. In line with this and to push the reasoning further, the authors also stipulated that HU treatment may improve the outcome of sickle cell pregnancy, without any teratogenic risk if started after the first trimester [26]—while preclinical animal studies or in vitro cellular data instead indicated that HU exposure may have some detrimental effects, including teratogenic effects, on the fetus.

In this study, our results showed that HU was stopped at least 15 days before conception in only 12% of pregnancies. In addition, 65.7% of women did not use any method of contraception, thereby confirming the high rate of unplanned pregnancies already described in the published literature of SCD population [27]. Importantly, the rate of pregnancy outcomes that we observed did not appear to differ following HU exposure and was similarly not affected by the period of HU exposure during pregnancy. Our results are in line with those observed from a French cohort of 128 women with SCD (95 with HbSS and 33 HbSC) and with the more recent study of Oakley et al., in which sickle cell disease patients received prophylactic transfusions, most without any HU exposure [28,29]. However, the small number of pregnancies without any HU exposure noted in our study does not allow for any statistical comparison between groups of pregnancies with versus without HU exposure. In 89 (81%) of the exposed pregnancies, HU was stopped during the first trimester, presumably as soon as the pregnancy was known. HU was also introduced or resumed in seven pregnancies and continued throughout pregnancy in four other women, thus underscoring the difficulty medical teams face in managing both a pregnancy in women with severe underlying disease/transfusion difficulties and the recommendation to stop HU treatment several months before conception. In the ESCORT-HU study, 64% of pregnancies resulted in a live birth regardless of HU exposure, which is higher than the 42% of pregnancies described in the MSH study, but similar to that reported in the Cooperative Study of Sickle Cell Disease (CSSCD) study (64%) [30]. In addition, without taking into account elective abortions, the percentage of live births increased up to 75%.

Concerning the limitations identified in this study, the cohort follow-up was not primarily undertaken with the objective of studying pregnant women. Because the state of our current data collection does not allow for an accurate detailing of related complications nor treatment that was received during the pregnancies, we did not want to compare them, even with results that emanate from previously published studies. It remains possible that some patients did not report pregnancies, but given the number of visits, this seems unlikely, since patients had a median visit every 5 months. However, we do consider these relevant data of enough interest to be shared with the scientific community.

## 5. Conclusions

The data on pregnancy outcomes after HU exposure in patients are rather reassuring when compared to the risks of maternal and perinatal complications related to sickle cell disease itself. Based on the importance of determining the actual risks and benefits of continuing HU treatment as a disease-modifying therapy during pregnancy in order to protect both mothers and babies, our study importantly revealed that in 89 (81%) of the exposed pregnancies during the first trimester, no specific problem in newborns nor maternal death could be reported. The percentage of live births from our ESCORT-HU study was 75%, without taking into account elective abortions, and the preterm delivery rate (16%) was similar to that previously reported in HbSS patients (16%) [28,29]. This is one of the few studies showing the frequency of miscarriage in sickle cell patients. Notably, the fertility of the SCD women treated with HU did not appear to be altered by the treatment nor related to it. However, given the insufficient number of patients, we cannot draw firm conclusions about the safety of the treatment. The data concerning complications, TF, and caesarean sections in these women are not exhaustive, because the cohort follow-up was not undertaken with this objective in mind, but these data seem relevant enough to be shared. Although these clinical observations are promising, pregnancy in HU-treated women still requires special management and vigilance by a multidisciplinary team, and also a strict follow-up. An earlier planned pregnancy follow-up would be even more adequate and important for women treated with HU [25,31,32]. In spite of the therapeutic indication of HU, which presages more expressive pathologies in women with sickle cell disease, in our study there was no maternal mortality. Finally, further studies are still necessary to fully document the risks/benefits of HU during pregnancy both for mothers and newborns. Information on all pregnancies in SCD patients treated with HU currently continue to be collected in the ongoing ESCORT-HU Extension study, regardless of HU exposure: initiation, discontinuation, and/or possible resumption during pregnancy.

## Figures and Tables

**Figure 1 biomedicines-11-00597-f001:**
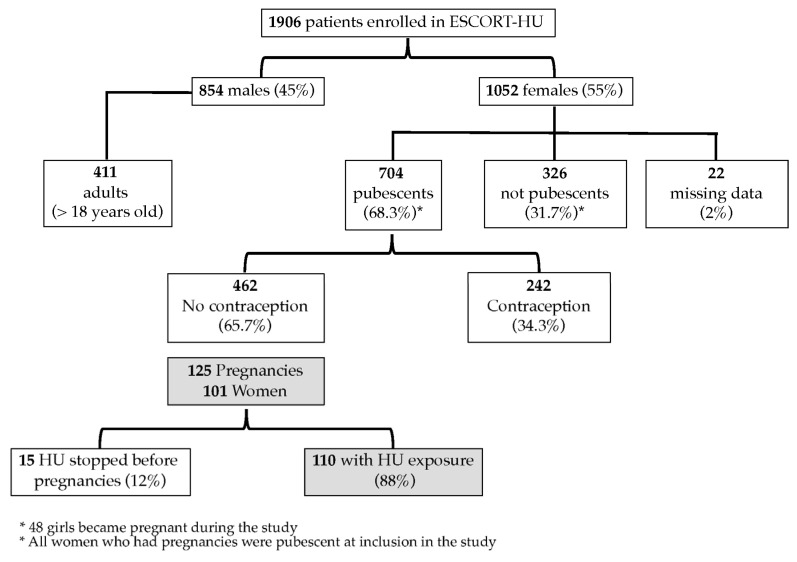
Clinical trial flowchart.

**Table 1 biomedicines-11-00597-t001:** Number of pregnancies before and during the study.

Number of Pregnancies during ESCORT-HU	125	(%)
Women with at least 1 pregnancy	101	81%
Women with 2 pregnancies	19	15%
Women with 3 pregnancies	5	4%
Number of twin pregnancies	3	2%
**Women with at least a pregancy before enrollment (368/691)**		54.9%
**Number of children before enrollment**	0	27 (7.4%)
	1	154 (42%)
	2	120 (32.9%)
	3	46 (12.6%)
	4	18 (4.9%)
	Not Reported	3

**Table 2 biomedicines-11-00597-t002:** Characteristics of pregnant women (* VOC 2.5 + 1.4/years).

Women Characteristics:	Pregancy N(%)	Global ESCORT HU Population
**Genotype** SS	89 (88%)	84.0%
Sβ0	6 (5.9%)	6.0%
Sβ+	4 (4.0%)	7.0%
SC	1 (0.1%)	2.0%
ND	1 (0.1%)	0.2%
**HU therapeutic indication**	**N**	
VOC *	66	
ACS	23	
anemia	6	
Organopathy/vasculopathy	7	
Not reported	2	
**Women characteristics:**	**Median [IQR1-3]**	
Age of initiation of Siklos (years)	28 [24–32]	
Siklos Dose at initiation (mg/kg)	15 [13–19]	
Age of first menstrual period (years)	15 [13–19]	
Age of first pregnancy (years)	30 [26–33]	
Follow–up duration (month)	52 [37–60]	
Number of visits	10 [8–12]	
Age of pregnancy in the study (years)	31 [26–35]	
Hb (g/dl)	8.8 [7.9–9.5]	
HbF (%)	8.8 [4.9–18.4]	

**Table 3 biomedicines-11-00597-t003:** Exposure periods to HU during pregnancies (* LMP: Last menstrual period).

HU Exposure	Number	%
No SIKLOS exposure	15	12%
≤4 weeks post-LMP *	22	17%
HU stopped between 4 and 12 weeks post-LMP	4	3%
≤12 weeks post-LMP	69	54%
≤20 weeks post-LMP	8	6%
Intro >12 weeks post-LMP	3	2%
HU throughout pregnancy	4	3%
Unknown	3	2%

**Table 4 biomedicines-11-00597-t004:** Outcomes of pregnancies in ESCORT-HU SCD women following HU exposure or not.

Outcome of Pregnancies	HU Exposure *n* = 110	No HU Exposure (*n* = 15)
Normal birth (≥36SA)	50 (45%)	6 (40%)
Premature birth (<36SA)	19 (17%)	1 (6%)
Voluntary abortion	18 (16%)	
Miscarriage	14 (13%)	3 (20%)
Ectopic pregnancy	1 (1%)	
Live birth (unspecified number of SA)	1 (1%)	5 (33%)
Anembryonic gestation	1 (1%)	
Stillbirth	1 (1%)	
Abortion for medical reason	1 (1%)	
Unknown	7 (6%)	

**Table 5 biomedicines-11-00597-t005:** Maternal complications in ESCORT-HU SCD women following HU exposure or not.

Maternal Complications	HU Exposure (%)	Without HU Exposure (%)
	*N* = 110 Pregnancies in 91 Women	*N* = 15 Pregnancies in 10 Women
**Related to SCD**		
VOC	25 (22.7%)	1 (6.7%)
Acute anemia	5 (4.5%)	
Cholestasis	3 (2.7%)	
Maternal cardiomyopathy	2 (1.8%)	
Pulmonary embolism	1 (0.9%)	
Splenic sequestration	1 (0.9%)	
Unspecified hospitalization	Not reported	2 (13.3%)
**Related to the pregnancy**		
Eclampsia/pre-eclampsia/hypertension	10 [2/5/3] (9.1%)	
Wall hematoma	2 (1.8%)	
Premature rupture of membranes	2 (1.8%)	
Post-delivery bleeding	2 (1.8%)	
Gestational diabetes	2 (1.8%)	
**Iatrogenic complications**		
Infection (catheter-related, unspecified)	2 (1.8%)	
Femoral catheter thrombosis	1 (0.9%)	
Morphine addiction	1 (0.9%)	
**Total of maternal complications reported**	**60 [54.5%]**	**3 [20%]**

## Data Availability

Not applicable.

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
