# Peer review of "Outcomes of Pregnancy in Sickle Cell Disease Patients: Results from the Prospective ESCORT-HU Cohort Study"

_biomedicines, 2023, doi:10.3390/biomedicines11020597_

Round 1

Reviewer 1 Report

Materials and Methods

Line 58: This is a cohort study and it should be reported in the methodology description

Lines 66-69: “As in the Multicenter Study of Hydroxyurea in SCD (MSH) clinical trial, no statistical testing was performed since the study was not specifically designed to assess the safety of HU during pregnancy in SCD women and data were collected from routine visit of the patients to hospital “  This statement does not necessarily justify the absence of any statistical analysis. Kroner et al have used the odds ratio method and multivariate analysis to present their data. The authors’ comment may be more suitable to report a low statistical power rather than the inability to perform any statistical analysis.  Please elaborate

Lack of reporting recruitment criteria

A more detailed description of the study population is required. Age, SCD type (homozygosity, double heterozygosity), inclusion and exclusion criteria. Reporting the SCD type is really important considering that the risk of complications in different in each patient subgroup.  Please elaborate

In general, the material and methods section is too short.

Lines 87-89: Please comment on why is it important to report the mean duration of HU treatment regarding the pregnancy outcome in SCD patients. If there is indeed important to report the HU treatment duration, it should be better reported in patient-years, considering that not all patients were recruited at the same time point. Please elaborate and correct

Figure 1 FLOWCHART

The data should be presented uniformly. Percentages are missing in some boxes. Please correct

The clinical trial flowchart is misleading. It should be made clear, how many were the previous pregnancies (before enrolment) and the number of pregnancies following enrolment. Please correct

RESULTS

Are there any data regarding the outcome of pregnancies before enrolment? It would be helpful to report whether there was any previous obstetric history. Please comment

In both tables 1 and 2, it is difficult to understand and clearly interpret the content. I suggest presenting the results using the odds ratio method. Please correct

A multivariate analysis is recommended, in order to clarify whether HU is or not an independent risk factor. 

The data reported in the text are difficult to read. I suggest reporting the data in a form of a table or figure legend.

Male patients should also discontinue HU in terms of the possible harmful effects on the sperm. It would be interesting to report any available data regarding the effects of HU on the sperm of the participating male patients and the pregnancy outcomes of their partners. Please comment

It is of great clinical significance to report whether pregnancies without HU exposure involved women who did or did not receive HU systematically. In Table 2 it is striking that pregnancies without HU exposure had almost no serious complications. This difference could be attributed to a selected subgroup of SCD patients with no strong indications for HU treatment. Please elaborate

Conclusions

The authors should report the limitations of the study (statistical power, selection bias, follow up etc)

Author Response

Reviewer 1

Materials and Methods:

Line 58: This is a cohort study and it should be reported in the methodology description.

We first would like to thank the reviewer 1 for his relevant comments and suggestions all over the manuscript  that would undoubtedly help in enhancing clarity of our submitted work. This informations was added as seen line 61: “Patients were enrolled in the study cohort if they satisfied all of the following criteria: male or female ambulatory patients, aged 2 years and more, with symptomatic SCD, entitled to be treated (patients enrolled under protocol Amendment 1) or treated (patients enrolled under protocol Amendment 2) with HU tablets, having been informed of the study by the initiating physician and consenting to participate to the cohort (for children, the persons having parental authority were informed and gave participation consent).”

Lines 66-69: “As in the Multicenter Study of Hydroxyurea in SCD (MSH) clinical trial, no statistical testing was performed since the study was not specifically designed to assess the safety of HU during pregnancy in SCD women and data were collected from routine visit of the patients to hospital “This statement does not necessarily justify the absence of any statistical analysis. Kroner et al have used the odds ratio method and multivariate analysis to present their data. The authors’ comment may be more suitable to report a low statistical power rather than the inability to perform any statistical analysis.  Please elaborate

Given the secondary nature of this study, we do not have sufficiently complete data to draw any significant nor relevant comparisons. Indeed, the e-CRF did not collect data on pregnancy and its related complications. People in charge to collect the data were indeed on several sites between maternities, sickle cell centers and transfusion centers, we thus regret that some of these data are missing especially those concerning TF, alloimmunizations or cesarean sections. We however slightly modified this section as you will see in our manuscript.

Lack of reporting recruitment criteria

A more detailed description of the study population is required. Age, SCD type (homozygosity, double heterozygosity), inclusion and exclusion criteria. Reporting the SCD type is really important considering that the risk of complications in different in each patient subgroup.  Please elaborate

We totally agree with the reviewer’s suggestions and have made proper modifications in our manuscript to include a table of genotypes and patient characteristics. New Tables were indeed added in our main text manuscript to enhance clarity as suggested by the reviewer (for a total of 5 tables).

In general, the material and methods section is too short.

We fully agree with the reviewer's comments and have in consequence modified the manuscript text to include the criteria of ESCORT-HU, from line 59 to 70: “ESCORT-HU was a multicentric, prospective, non-interventional, European cohort study initiated to collect information on long-term safety of HU tablets when used in current practice. Patients were enrolled in the study cohort if they satisfied all of the following criteria: male or female ambulatory patients, aged 2 years and more, with symptomatic SCD, entitled to be treated (patients enrolled under protocol Amendment 1) or treated (patients enrolled under protocol Amendment 2) with HU tablets, having been informed of the study by the initiating physician and consenting to participate to the cohort (for children, the persons having parental authority were informed and gave participation consent). No exclusion criteria were defined in the study. The population have given their consent or non-opposition (depending on local regulations) to participate in the study. This study was approved by National Ethics Committees and lasted for 10 years from January 2009 to March 2019.”

Lines 87-89: Please comment on why is it important to report the mean duration of HU treatment regarding the pregnancy outcome in SCD patients. If there is indeed important to report the HU treatment duration, it should be better reported in patient-years, considering that not all patients were recruited at the same time point. Please elaborate and correct

We agree with the reviewer in some extent, but however, we had data on Siklos and some of these patients already received another form of hydroxyurea and we are missing this information. Notably, this information was important to highlight the fact that being on HU treatment did not seem to affect fertility.

Figure 1 FLOWCHART

The data should be presented uniformly. Percentages are missing in some boxes. Please correct

This has been corrected in the new submitted Figure as you will see.

The clinical trial flowchart is misleading. It should be made clear, how many were the previous pregnancies (before enrolment) and the number of pregnancies following enrolment. Please correct

We agree with the reviewer and submitted new tables in our main manuscript to clarify this important point as you will see.

RESULTS

Are there any data regarding the outcome of pregnancies before enrolment? It would be helpful to report whether there was any previous obstetric history. Please comment

We are aware that we may not have the total number of pregnancies, as some data including miscarriage histories have not been recorded in the CRF. To enhance clarity and try to reply to this relevant comment, we added a table with all the collected or available data.

In both tables 1 and 2, it is difficult to understand and clearly interpret the content. I suggest presenting the results using the odds ratio method. Please correct

We agree with the reviewer's comments and have modified it as you will see in our main manuscript.

The data reported in the text are difficult to read. I suggest reporting the data in a form of a table or figure legend.

As aforementioned, we included more tables to enhance clarity in our main new submitted manuscript.

Male patients should also discontinue HU in terms of the possible harmful effects on the sperm. It would be interesting to report any available data regarding the effects of HU on the sperm of the participating male patients and the pregnancy outcomes of their partners. Please comment

We agree with the reviewer’s suggestion but we do not have any exhaustive data in men.

It is of great clinical significance to report whether pregnancies without HU exposure involved women who did or did not receive HU systematically. In Table 2 it is striking that pregnancies without HU exposure had almost no serious complications. This difference could be attributed to a selected subgroup of SCD patients with no strong indications for HU treatment. Please elaborate

We thank the reviewer for his relevant comment but there were only 15 patients in the unexposed group and this number remains too small to draw any significant conclusions.

Conclusions

The authors should report the limitations of the study (statistical power, selection bias, follow up etc …)

We agree with the reviewer's comments and have modified the text as you will see lines 227 and 243: “Concerning the limitations of this study, the cohort follow-up was not primarily done with the objective of studying pregnant women. Because the state of our current data collection does not allow for accurate detailing of related complications nor treatment received during the pregnancies, we therefore did not want to compare them even with results that emanate from previous publications. However, we do consider these relevant data of enough interest to be shared with the scientific community.” and The data concerning complications, TF, caesarean sections in these women are not ex-haustive because the cohort follow-up was not done with this objective in mind, but these data seem relevant enough to be shared.”

Reviewer 2 Report

This manuscript reports on the outcomes of pregnancies observed in the ESCORT-HU trial, a phase IV observational study on the use of hydroxyurea (HU) in sickle cell disease (SCD).

Such observations are important for physicians taking care of patients with SCD and may advise future treatment guidelines. However, this report lacks originality because the main data that are presented, the number and outcomes of pregnencies, were reported previously by the same group (de Montalembert et al. 2021, PMID 34224583): "Overall, 125 pregnancies were reported for 101 women: 110 withexposure to HU (generally during the first trimester) and 15 withoutHU exposure (HU stopped at least 15 days before conception),despite the recommendation to discontinue HU 3 to 6 months beforeconception. The pregnancy outcomes included live births (n=77),elective termination (n=18), spontaneous abortion (n=17), or on-going pregnancy (n=9). Other reported outcomes were anembryonicgestation (n=1), ectopic pregnancy (n=1), and termination for medi-cal reasons (n=2). No malformations among the neonates or maternal deaths were reported."

Unfortunately, the authors fail to cite their important previous work that does not only anticipate the main results, but in addition describes the ESCORT-HU cohort in more details than the current manuscript does.

I suggest the following revisions to the manuscript:

1. Cite the previous work and present additional data that clearly provide new results. These should include

- a table with a more detailed description of the patients (genotype, age at first pregnancy, age at current pregnancy, history of VOC, history of transfusions, number of previous pregnancies)

- a more detailed description of complications and treatments during pregnancy. How many patients received regular transfusion during pregnancy, how many episodic transfusions, which were indications for transfusions? 

- was there any correlation between complications, pregnancy outcome and patient characteristics, e.g. genotype?

2. The flowchart (fig 1) implies that only patients who were pubescent at enrollment became pregnant. Please clarify if any patients who were pre-pubertal at enrollment but became pubescent during the 10 years follow up were included. 

3. "Biological parameters" were available for a small number of patients with pregnancies. Please discuss if the lack of detailed data for a large proportion of patients indicates that there is a bias, e.g. towards under-reporting of complications. 

4. Please discuss other limitations of the trial. These may include incomplete data, a recruitment bias, incomplete reporting of AE and SAE. If, e.g., "none of these events was suspected to be related to HU", clarify who assessed causality how.

5. Compare the pregnancy outcomes in the ESCORT-HU cohort to that of other patients with SCD and cite additional references that provide an estimate for the "background complication rate" during pregnancy in patients with SCD, e.g. Oakley et al. BJH 2021, PMID 34881428.

Author Response

Reviewer 2

I suggest the following revisions to the manuscript:

  1. Cite the previous work and present additional data that clearly provide new results. These should include

- a table with a more detailed description of the patients (genotype, age at first pregnancy, age at current pregnancy, history of VOC, history of transfusions, number of previous pregnancies)

We first would like to thank the reviewer 2 for his relevant comments and suggestions. We have taken into account the reviewer's remarks by including our references and we also added a table with detailed informations and patient characteristics.

- a more detailed description of complications and treatments during pregnancy. How many patients received regular transfusion during pregnancy, how many episodic transfusions, which were indications for transfusions?

We thank the reviewer and we agree that these questions are very relevant. Unfortunately, we do not have the answers. Transfusions and deliveries indeed take place in various departments and this information could not be found in the e-CRF. In addition, there were no additional visit to have this data. While these displayed data may not be the most exhaustive data, we do believe that these informations are very important for the community and deserve to be shared.

  1. The flowchart (fig 1) implies that only patients who were pubescent at enrolment became pregnant. Please clarify if any patients who were pre-pubertal at enrolment but became pubescent during the 10 years follow up were included.

Thanks for this comment. This was added in the manuscript and all pregnancies during the follow-up were included.

  1. "Biological parameters" were available for a small number of patients with pregnancies. Please discuss if the lack of detailed data for a large proportion of patients indicates that there is a bias, e.g. towards under-reporting of complications.

This was added in the manuscript.

  1. Please discuss other limitations of the trial. These may include incomplete data, a recruitment bias, incomplete reporting of AE and SAE. If, e.g., "none of these events was suspected to be related to HU", clarify who assessed causality how.

This was modified in the Conclusion part.

  1. Compare the pregnancy outcomes in the ESCORT-HU cohort to that of other patients with SCD and cite additional references that provide an estimate for the "background complication rate" during pregnancy in patients with SCD, e.g. Oakley et al. BJH 2021, PMID 34881428.

We thank the reviewer for providing us other references. We have included these 2 relevant references in our manuscript and made a prompt discussion/comparison as asked, line 210: “Pregnancy outcomes did not appear to differ following HU exposure and was similarly not affected by the period of HU exposure during pregnancy. Our results are similar to those observed from a French cohort of 128 women with SCD (95 with HbSS and 33 HbSC) and to the more recent study of Oakley et al. receiving prophylactic transfusions and without any HU exposure [16,17].”

Round 2

Reviewer 1 Report

No further comments or suggestions

Author Response

No specific comments have been asked by the Reviewer 1 during the second round of revisions. We are very grateful for his relevant suggestions that helped us to enhance clarity of your submitted work. 

Reviewer 2

The authors have provided additional information, most importantly a more detailed description of the methods and table 2 that shows the characteristics of pregnant women with SCD.The authors did not clearly mark the changes they made in the manuscript and waived a detailed point by point response.

We warmly thank the reviewer 2 for his comments and have now provided a new version where Track Changes have been activated to enhance clarity of our submitted work.

  1. Flowchart 1 still implies, intentionally or not, that only women who were pubescent at study entry and who did not take contraception became pregnant. If this were true it would be remarkable. More likely, some women who became pubescent during the study period and who did take contraception at study entry became pregnant. I suggest to delete the three boxes at the bottom because these numbers (15 pregnancies without HU exposure, 110 with HU exposure) are also given in the text. Instead, I suggest to assign to each of the boxes "pubescents", "not pubescents", "no contraception", "contraception" the number and proportion of women who became pregnant. If these data are not available, it should be clearly stated in the text.

We thank the reviewer 2 for his suggestion. The pubescents or not pubescents status is the status determined at the enrollment, 48 girls reached puberty during the study but none of them became pregnant. Thus, to enhance clarity, we added this information to both in the flowchart AND in the text manuscript line 108: “Forty eight girls reached puberty during the study but none of them became pregnant.”

  1. The authors mention now in the very last three sentences of the discussion that the study may have some limitations. I believe this is not sufficient. Important limitations are not mentioned but need to be discussed: - reporting bias: I assume that most pregnancies will have been reported. However, miscarriages or voluntary abortions may not have been reported to the hematologist and thus not be documented. Is there any estimate of the number of miscarriages that were not reported? An important number in this respect would be the frequency of "routine visits" that generated registry data. If a patient was followed once per year, a miscarriage goes undetected more easily than if the patient comes every month.

As rightly stated by the reviewer 2, we wanted to draw his attention to the fact that there were no mandatory visits for the study and that visits were only based on those scheduled by clinicians as part of their routine practice. We however added this information in the Materials and Methods section to enhance clarity. However, pregnancy information was required in the eCRF and we had only 21 missing pregnancy data. Patients had a median of 10 [8-12] visits during their median inclusion period of 52.2 [37.2 - 60.1] months, which seems to us reasonable in terms of information collection.

- Recruitment bias: e.g., patients with homozygous SCD are more likely to be treated with HU than e.g. SCD-S/C. The patient characteristics in Table 2 must viewed in this light. I suggest that the authors add the corresponding numbers for the complete ESCORT-HU cohort for a comparison and discuss a possible recruitment bias.

Siklos is not often used for SC patients and the majority of patients are SS homozygous or Sβ0 thalassemic. According to the Reviewer 2 suggestion, we added in Table 2 the overall % of each genotype in the total Escort HU cohort population.

- Missing data: The authors state very generally "follow-up was not done with the objective of studying pregnant women". It would be much more helpful to quantitate the proportion of missing data.

The reviewer 2 is indeed correct and we are now prodiving detailed information. In the whole study ESCORT-HU, 51 patients (2.6%) were lost during the follow-up. While patients came regularly for consultation on HU, these visits were spaced out during pregnancy due to the cessation of HU and pregnancy follow-up visits. However, in the eCRF, data regarding the occurrence of pregnancy were requested and investigators had to address this question. These results on pregnancies, miscarriages or abortions were therefore completely reliable, but the data related to pregnancy follow-up were however not included in the eCRF. In conclusion, missing data related to pregnancies are thus limited at 21.

- How many patients were lost to follow up during the study? What was the median time until loss to follow up? How many of the eCRF items were left open? The Methods section states "data...were monitored in accordance with the protocol". Please specify what monitoring was done and discuss how this monitoring could or could not prevent missing data.

Extensive monitoring was not foreseen at the initiation of the study. However, quality control of the data was performed since 2012 (added information in the text manuscript line 77). For a given site, the first on-site quality control visit was triggered after inclusion of the 5th patient at this specific site (i.e. 8 patient visits available in the eCRF with at least the first year of follow-up for one of the patient, meaning 3 visits). After the first on-site visit, only the 16 biggest recruiting sites were visited yearly. Furthermore, 5% of the data at each site were controlled. A total of 142 quality control visits were performed during the study by the CRO trained staff: including 102 in France, 23 in French overseas territories and departments, 11 in Germany, 3 in Greece and finally 3 in Italy. Importantly, after each visit, results were sent to the sites in order to complete or correct the data. No site closure nor major action regarding sites were needed during the study.

- Assessment of causality: Line 171/172: "none of these events were suspected to be related to HU": Who assessed causality?

We thank the reviewer for this relevant question. The committee of clinical investigators carefully studied these files and concluded or not on the immutability of the anomalies according to exposure period, exposure duration and the nature of the anomalies. This information has been added in our manuscript line 183: “The committee of clinical investigators studied these files and concluded on the imput-ability or not of the anomalies according to the period of exposure, duration of exposure and the nature of the anomalies.”

  1. Line 136/137: "no malformations were reported", line 169:"1 scimitar syndrome, 1 fetal arrhythmia, 1 cardic disorder, 1 patent ductus arteriosus". Please reconcile. At least scimitar syndrom should be classified as malformation.

The reviewer is absolutely right and this is a mistake from us as this syndrome is considered a non-major malformation, but it still is a malformation. To correct this and be accurate, we completely deleted this sentence from line 136.

 To go further, we can further explain about malformations: In July 2016, the mother became pregnant (last menstrual period dated July 7, 2016) and the treatment with SIKLOS was stopped in July 2016. The patient received it only a few days.  This is a malformation with a frequency of 1/10000 to 1/33000. Malformations of this nature are not due to a period of teratogenicity in the first trimester and this woman had less than one month of exposure to HU alone. The committee of investigators after studying this case considered that it was not attributable to HU.

My co-authors and I sincerely hope that the modifications made will enable you to consider our revised work for publication in Biomedicines. Should you require any additional information, please do not hesitate to contact me at: anoosha.habibi@aphp.fr

Sincerely,

Anoosha Habibi, MD

Reviewer 2 Report

The authors have provided additional information, most importantly a more detailed description of the methods and table 2 that shows the characteristics of pregnant women with SCD.

Unfortunately, the authors did not clearly mark the changes they made in the manuscript and waived a detailed point by point response. 

I have the following comments and suggestions:

1. Flowchart 1 still implies, intentionally or not, that only women who were pubescent at study entry and who did not take contraception became pregnant. If this were true it would be remarkable. More likely, some women who became pubescent during the study period and who did take contraception at study entry became pregnant. I suggest to delete the three boxes at the bottom because these numbers (15 pregnancies without HU exposure, 110 with HU exposure) are also given in the text. Insted, I suggest to assign to each of the boxes "pubescents", "not pubescents", "no contraception", "contraception" the number and proportion of women who became pregnant. If these data are not available, it should be clearly stated in the text.

2. The authors mention now in the very last three sentences of the discussion that the study may have some limitations. I believe this is not sufficient. Important limitations are not mentioned but need to be discussed: - reporting bias: I assume that most pregnancies will have been reported. However, miscarriages or voluntary abortions may not have been reported to the hematologist and thus not be documented. Is there any estimate of the number of miscarriages that were not reported? An important number in this respect would be the frequency of "routine visits" that generated registry data. If a patient was followed once per year, a miscarriage goes undetected more easily than if the patient comes every month.

- recruitment bias: e.g., patients with homozygous SCD are more likely to be treated with HU than e.g. SCD-S/C. The patient characteristics in Tab 2 must viewed in this light. I suggest that the authors add the corresponding numbers for the complete ESCORT-HU cohort for a comparison and discuss a possible recruitment bias. 

- missing data: The authors state very generally "follow-up was not done with the objective of studying pregnant women". It would be much more helpful to quantitate the proportion of missing data. How many patients were lost to follow up during the study? What was the median time until loss to follow up? How many of the eCRF items were left open? The Methods section states "data...were monitored in accordance with the protocol". Please specify what monitoring was done and discuss how this monitoring could or could not prevent missing data.

- assessment of causality: Line 171/172: "none of these events were suspected to be related to HU": Who assessed causality?

3. Line 136/137: "no malformations were reported", line 169:"1 scimitar syndrome, 1 fetal arrhythmia, 1 cardic disorder, 1 patent ductus arteriosus". Please reconcile. At least scimitar syndrom should be classified as malformation.

Author Response

Reviewer 2

The authors have provided additional information, most importantly a more detailed description of the methods and table 2 that shows the characteristics of pregnant women with SCD.The authors did not clearly mark the changes they made in the manuscript and waived a detailed point by point response.

We warmly thank the reviewer 2 for his comments and have now provided a new version where Track Changes have been activated to enhance clarity of our submitted work.

  1. Flowchart 1 still implies, intentionally or not, that only women who were pubescent at study entry and who did not take contraception became pregnant. If this were true it would be remarkable. More likely, some women who became pubescent during the study period and who did take contraception at study entry became pregnant. I suggest to delete the three boxes at the bottom because these numbers (15 pregnancies without HU exposure, 110 with HU exposure) are also given in the text. Instead, I suggest to assign to each of the boxes "pubescents", "not pubescents", "no contraception", "contraception" the number and proportion of women who became pregnant. If these data are not available, it should be clearly stated in the text.

We thank the reviewer 2 for his suggestion. The pubescents or not pubescents status is the status determined at the enrollment, 48 girls reached puberty during the study but none of them became pregnant. Thus, to enhance clarity, we added this information to both in the flowchart AND in the text manuscript line 108: “Forty eight girls reached puberty during the study but none of them became pregnant.”

  1. The authors mention now in the very last three sentences of the discussion that the study may have some limitations. I believe this is not sufficient. Important limitations are not mentioned but need to be discussed: - reporting bias: I assume that most pregnancies will have been reported. However, miscarriages or voluntary abortions may not have been reported to the hematologist and thus not be documented. Is there any estimate of the number of miscarriages that were not reported? An important number in this respect would be the frequency of "routine visits" that generated registry data. If a patient was followed once per year, a miscarriage goes undetected more easily than if the patient comes every month.

As rightly stated by the reviewer 2, we wanted to draw his attention to the fact that there were no mandatory visits for the study and that visits were only based on those scheduled by clinicians as part of their routine practice. We however added this information in the Materials and Methods section to enhance clarity. However, pregnancy information was required in the eCRF and we had only 21 missing pregnancy data. Patients had a median of 10 [8-12] visits during their median inclusion period of 52.2 [37.2 - 60.1] months, which seems to us reasonable in terms of information collection.

- Recruitment bias: e.g., patients with homozygous SCD are more likely to be treated with HU than e.g. SCD-S/C. The patient characteristics in Table 2 must viewed in this light. I suggest that the authors add the corresponding numbers for the complete ESCORT-HU cohort for a comparison and discuss a possible recruitment bias.

Siklos is not often used for SC patients and the majority of patients are SS homozygous or Sβ0 thalassemic. According to the Reviewer 2 suggestion, we added in Table 2 the overall % of each genotype in the total Escort HU cohort population.

- Missing data: The authors state very generally "follow-up was not done with the objective of studying pregnant women". It would be much more helpful to quantitate the proportion of missing data.

The reviewer 2 is indeed correct and we are now prodiving detailed information. In the whole study ESCORT-HU, 51 patients (2.6%) were lost during the follow-up. While patients came regularly for consultation on HU, these visits were spaced out during pregnancy due to the cessation of HU and pregnancy follow-up visits. However, in the eCRF, data regarding the occurrence of pregnancy were requested and investigators had to address this question. These results on pregnancies, miscarriages or abortions were therefore completely reliable, but the data related to pregnancy follow-up were however not included in the eCRF. In conclusion, missing data related to pregnancies are thus limited at 21.

- How many patients were lost to follow up during the study? What was the median time until loss to follow up? How many of the eCRF items were left open? The Methods section states "data...were monitored in accordance with the protocol". Please specify what monitoring was done and discuss how this monitoring could or could not prevent missing data.

Extensive monitoring was not foreseen at the initiation of the study. However, quality control of the data was performed since 2012 (added information in the text manuscript line 77). For a given site, the first on-site quality control visit was triggered after inclusion of the 5th patient at this specific site (i.e. 8 patient visits available in the eCRF with at least the first year of follow-up for one of the patient, meaning 3 visits). After the first on-site visit, only the 16 biggest recruiting sites were visited yearly. Furthermore, 5% of the data at each site were controlled. A total of 142 quality control visits were performed during the study by the CRO trained staff: including 102 in France, 23 in French overseas territories and departments, 11 in Germany, 3 in Greece and finally 3 in Italy. Importantly, after each visit, results were sent to the sites in order to complete or correct the data. No site closure nor major action regarding sites were needed during the study.

- Assessment of causality: Line 171/172: "none of these events were suspected to be related to HU": Who assessed causality?

We thank the reviewer for this relevant question. The committee of clinical investigators carefully studied these files and concluded or not on the immutability of the anomalies according to exposure period, exposure duration and the nature of the anomalies. This information has been added in our manuscript line 183: “The committee of clinical investigators studied these files and concluded on the imput-ability or not of the anomalies according to the period of exposure, duration of exposure and the nature of the anomalies.”

  1. Line 136/137: "no malformations were reported", line 169:"1 scimitar syndrome, 1 fetal arrhythmia, 1 cardic disorder, 1 patent ductus arteriosus". Please reconcile. At least scimitar syndrom should be classified as malformation.

The reviewer is absolutely right and this is a mistake from us as this syndrome is considered a non-major malformation, but it still is a malformation. To correct this and be accurate, we completely deleted this sentence from line 136.

 To go further, we can further explain about malformations: In July 2016, the mother became pregnant (last menstrual period dated July 7, 2016) and the treatment with SIKLOS was stopped in July 2016. The patient received it only a few days.  This is a malformation with a frequency of 1/10000 to 1/33000. Malformations of this nature are not due to a period of teratogenicity in the first trimester and this woman had less than one month of exposure to HU alone. The committee of investigators after studying this case considered that it was not attributable to HU.

My co-authors and I sincerely hope that the modifications made will enable you to consider our revised work for publication in Biomedicines. Should you require any additional information, please do not hesitate to contact me at: anoosha.habibi@aphp.fr

Sincerely,

Anoosha Habibi, MD

Round 3

Reviewer 2 Report

I thank the authors for adopting the reviewer's suggestions.

Regarding Fig 1 I have a minor comment: "*48 girls became pubescent during the study", not "*48 girls became pregnant during pregnancy"

Author Response

Reviewer 2

Regarding Fig 1 I have a minor comment: "*48 girls became pubescent during the study", not "*48 girls became pregnant during pregnancy".

We first would like to thank again the Reviewer 2 for having scrupulously read our manuscript and helped in improving it. This mistake was corrected in our new revised manuscript.

My co-authors and I sincerely hope that the modifications made will enable you to consider our revised work for publication in Biomedicines. Should you require any additional information, please do not hesitate to contact me at: anoosha.habibi@aphp.fr
